# A Proof of Principle Proteomic Study Detects Dystrophin in Human Plasma: Implications in DMD Diagnosis and Clinical Monitoring

**DOI:** 10.3390/ijms24065215

**Published:** 2023-03-08

**Authors:** Rachele Rossi, Camilla Johansson, Wendy Heywood, Heloise Vinette, Gabriella Jensen, Hanna Tegel, Albert Jiménez-Requena, Silvia Torelli, Cristina Al-Khalili Szigyarto, Alessandra Ferlini

**Affiliations:** 1The Dubowitz Neuromuscular Centre, UCL Great Ormond Street Institute of Child Health, London WC1N 1EH, UK; 2Medical Genetics Unit, Department of Medical Sciences, University of Ferrara, 44121 Ferrara, Italy; 3Department of Protein Science, KTH-Royal Institute of Technology, 11428 Stockholm, Sweden; 4Translational Mass Spectrometry Research Group, Genetics & Genomic Medicine Department, UCL Institute of Child Health, London WC1N 1EH, UK; 5Science for Life Laboratory, Department of Protein Science, KTH-Royal Institute of Technology, 11428 Stockholm, Sweden

**Keywords:** DMD, dystrophin protein, plasma assay, immunoassay, LC-MS/MS

## Abstract

Duchenne muscular dystrophy (DMD) is a rare neuromuscular disease caused by pathogenic variations in the *DMD* gene. There is a need for robust DMD biomarkers for diagnostic screening and to aid therapy monitoring. Creatine kinase, to date, is the only routinely used blood biomarker for DMD, although it lacks specificity and does not correlate with disease severity. To fill this critical gap, we present here novel data about dystrophin protein fragments detected in human plasma by a suspension bead immunoassay using two validated anti-dystrophin-specific antibodies. Using both antibodies, a reduction of the dystrophin signal is detected in a small cohort of plasma samples from DMD patients when compared to healthy controls, female carriers, and other neuromuscular diseases. We also demonstrate the detection of dystrophin protein by an antibody-independent method using targeted liquid chromatography mass spectrometry. This last assay detects three different dystrophin peptides in all healthy individuals analysed and supports our finding that dystrophin protein is detectable in plasma. The results of our proof-of-concept study encourage further studies in larger sample cohorts to investigate the value of dystrophin protein as a low invasive blood biomarker for diagnostic screening and clinical monitoring of DMD.

## 1. Introduction

Dystrophinopathies are progressive, X-linked muscular disorders characterised by progressive muscle degeneration and weakness. They include three main clinical phenotypes: the most severe Duchenne muscular dystrophy (DMD) (OMIM # 310200), the milder Becker muscular dystrophy (BMD) (OMIM # 300376), and the isolated dilated cardiomyopathy (OMIM # 302045) [1]. Dystrophinopathies are caused by mutations in the dystrophin gene (*DMD*) that leads to the absence of or heavy reduction of dystrophin protein (DYS). DYS is a large protein of 427 kDa, belonging to the spectrin protein superfamily, characterised by four distinct domains, providing a flexible link between the sarcomeric actin filaments and the sarcolemma. Among several isoforms, the full-length isoform Dp427m is predominantly expressed in skeletal and cardiac muscle [2]. Alterations in its function or expression result in a disorganisation of the dystrophin-associated glycoprotein complex that causes instability of the subsarcolemmal muscle membrane, making muscles susceptible to damage [3]. The time-to-diagnosis for DMD has hardly changed in the last two decades, despite the implementation of genetic diagnostic testing, with a constant interval of 2.5 years between the first signs observed and the definitive diagnosis. Patients receive diagnosis at an average age of 5 years, with high variability among different countries [4]. Early diagnosis in DMD boys represents a crucial step for allowing early treatments, optimal standard of care, and reduction of irreversible sequelae. New treatments aimed at partially restoring dystrophin protein translation (i.e., ataluren, eteplirsen, golodirsen) [5] have shown promising results especially when administered before a significant loss of muscle mass occurs. In addition, oncoming gene therapies might be more effective if provided early [6]. Consequently, there is a general consensus regarding the need of new non-invasive methods to screen and monitor DMD patients [7] using tools with a high diagnostic yield, repeatability, cost-effectiveness, and low invasiveness [8]. Testing for DYS has been performed in muscle tissues or in myogenic cells, which requires an invasive muscle biopsy procedure [9]. Recently, urinary stem cells (USCs) were proven to be very effective to study DMD mRNA, but are a poor resource for detecting DYS due to its extremely low level of expression [9]. Finally, thanks to the advances of next generation sequencing (NGS) techniques, genetic testing is now feasible and very accurate, with lower costs and high detection rate [10]. Nevertheless, having available a specific, low-invasive biochemical biomarker will be extremely valuable for supporting diagnosis of variants of unknown significance (VUS) [11]. The availability of a DYS-specific marker measurable in a low invasive way would be of great benefit to assay dystrophin expression for phenotype–genotype correlation, and to re-classify VUS [12].

Elevated creatine kinase (CK) levels in blood usually indicate skeletal muscle injury and for this reason it is the most commonly used serum biomarker for DMD. Unfortunately, CK lacks specificity since its levels increase in many forms of muscular dystrophy, physical exercise, and severe or acute infection in myositis [13]. Moreover, CK varies between days, and is not associated with DMD severity [14]. Notably, poor CK specificity was one of the main reasons why some DMD newborn screening, based on CK biochemical screening, was considered not accurate enough to be applied into health programs [15]. The CK test has improved its specificity and sensitivity and could be used for neonatal screening with results needing to be confirmed by genetic tests. However, it is not informative for disease or therapy monitoring [16]. Biomarkers are an unmet need also in DMD clinical trials either for monitoring available therapies or developing new ones, also considering that currently used standardised outcome measures include physical and imaging tests, which are either slow in delivering results or low throughput and costly [17]. The DYS protein assay in muscle biopsy is the only European Medicines Agency (EMA)-approved pharmacodynamic biomarker to monitor DMD-restoring therapies [18]. Dystrophin is normally expressed at low, and not constant, levels among individuals [19] and shows a lower rate of expression in DMD patients, making its detection extremely challenging [20], especially in fluids dominated by a high abundance of other proteins. Moreover, dystrophin is a protein with considerable dimension that localises to the cytoplasmic side of the sarcolemma. [21] and is not considered a constitutively expressed protein detectable in body fluids [22]. Recently, two dystrophin peptides of unknown origin, were found in dry blood spots of DMD patients and healthy controls, by targeted liquid chromatography mass spectrometry [23]. Possibly, the DMD fragments identified in blood spots may come from blood cells, i.e., granulocytes and B-cells, since these cells are known to express dystrophin, based on RNA-sequencing results [24]. Here, we aimed to detect the presence of dystrophin protein fragments in processed plasma, speculating that, like for titin and myoglobin [25], proteolytically cleaved fragments of skeletal muscle dystrophin could be released and exported, actively or passively, into the blood stream from affected tissues. Since DMD patients have pathogenic mutations that completely or partially abolish DYS expression, we did not expect to find full-length DYS in blood from these patients [26]. To investigate if any parts of DYS can be detected in plasma, we tested a suspension bead immunoassay with two anti-dystrophin-specific antibodies (Abs) against both the N-terminal and C-terminal domains of the DYS protein. In addition, we also used targeted liquid chromatography mass spectrometry (LC-MS/MS) method, which is independent of antibodies, to confirm DYS is detectable in plasma.

## 2. Results and Discussion

We designed an in-house suspension bead immunoassay as a proof-of-principle for investigating if DYS can be differentially detected in plasma from DMD patients compared to DMD carriers, healthy controls, and other neuromuscular disorders. Samples were tested at four different dilutions: 0.07%, 0.14%, 0.28%, and 0.56% of plasma to determine a detection range for the antibodies. This approach was selected because of the low volume of plasma needed for the assay and the availability of validated anti-DYS antibodies. Results showed that assays prepared at the highest concentration of plasma had the highest variation in non-specific background signal, negatively impacting reproducibility, as assessed through the coefficient of variation (%CV) of median fluorescent intensity (MFI) for positive and negative control beads. Therefore, to account for variation in assay preparation while maintaining low background, mean MFI was calculated across only the two lowest concentrations of plasma (two replicates at 0.07% and one replicate at 0.14%), as shown in Figure 1. The measurements had a coefficient of variation (CV) over replicates of around 10% for two of the three antibodies tested. HPA002725 had a CV of 8.9% (95% CI: 5.3% to 12.5%) over replicates and CA30121 had 11.2% (95% CI: 8.6% to 13.8%). In contrast, %CV over replicates for HPA023885 was 37.2% (95% CI: 31.8% to 42.7%). Mean MFI and the relative standard deviation, for each individual sample among the three antibodies and the empty beads assay, are shown in Appendix A. Relative abundances of N-terminal DYS in plasma, as detected by HPA002725 (Figure 1A), was significantly lower in DMD patients compared to other groups (*p*-value < 0.05, Wilcoxon rank sum test). In contrast, there was a significantly lower level of C-terminal DYS, as detected by HPA023885, in DMD compared to other neuromuscular diseases, but not when comparing to healthy controls (Figure 1B). One explanation for this could be that the assay conditions tested are outside the linear detection range for HPA023885, which is further suggested by the high MFI obtained, the lack of linear relationship between different dilution conditions and high %CV. MFI values observed in healthy controls are quite divergent for both antibodies anti-DMD. In a recent plasma protein analysis, comprising 94 donors with no obvious symptoms, it was shown that each individual has a unique protein profile distinct from the profiles of other individuals [27]. To solve this limit it is necessary to perform several experiments in order to assess a range of predictivity, and maybe using multiple Abs anti-DMD to increase the test specificity. Relative abundance of CA3 was used as a positive control in this experiment as it has previously been shown by us to be elevated in both serum and plasma from DMD patients compared to healthy controls [28]. The antibody against CA3 showed, as expected, a significantly increased relative abundance in the DMD group compared to other groups (Figure 1C). CA3 is known to be a deterioration marker for type 1 muscle fibres [29] and thanks to its limited tissue expression, it appears more specific and sensitive than CK [30]. It was also demonstrated that it accumulates in blood as a consequence of muscle damage [29] and, for this reason, we cannot consider it a specific marker for DMD although it was observed be able to discriminate between DMD and healthy controls and between DMD and BMD [28].

Before pursuing validation of the DYS abundance differences in larger cohorts, we aimed at confirming detection of the protein in healthy plasma by an antibody independent technique such as mass spectrometry. Considering the challenges in detecting low abundance proteins from tissue leakage in biological samples with a high protein dynamic range such as plasma, we decided to use targeted liquid chromatography triple quadrupole mass spectrometry. This technique is considered one of the main analytical methods in proteome research, and is suitable for protein/peptide identification in body fluids [31,32]. Due to technological advances, it is becoming increasingly used for biomarker quantification in clinical assays [33,34] and was recently used to detect DYS in dried blood spots [23]. Moreover, recently, a targeted proteomic test for SARS-CoV-2 has become the first targeted proteomic clinical test UKAS accredited and MHRA approved for clinical use in patients [34]. An initial LC-MS/MS experiment was performed to detect DYS in control plasma using the targeted proteomics approach with a commercial digestion standard MASSPrep yeast enolase, an established approach to control for digestion and LC-MS/MS analysis of digested proteins. Subsequently, a multiple reaction monitoring method was set up using recombinant peptides obtained digesting the D1-D2 PrESTs (labelled in grey and colour in Figure 2A and specified in Appendix A). Analysis of five control plasma samples showed that three tryptic peptides could be reliably detected (Figure 2B). Specifically detected were tryptic peptides YQS (YQSEFEEIEGR), LLV (LLVSDIQTIQPSLNSVNEGGQK), and TTE (TTENIPGGAEEISEVLDSLENLMR), respectively labelled in purple, turquoise, and yellow in Figure 2A. Abundance of the tryptic peptides in the samples is summarised in Figure 2C and indicates that the abundance pattern of the three tryptic peptides varies across samples.

Tryptic peptides LLV and TTE are the lowest abundance peptides (Figure 3A,B), and they share the same expression profile with the only exception that LLV is not detected in one sample (Figure 3A). In contrast, the most abundant YQS tryptic peptide showed a different expression profile than LLV and TTE, among subjects (Figure 3C). Transition information used for quantitation is given with the tryptic peptide sequence for each graph (Appendix A).

The lack of concordance observed between peptide abundance could be due to an individual sample effect as recently reported in the plasma proteomic study conduct by Tebani et al. [27]. It is also possible that post-translational or exporting modifications occur on these peptides, considering that plasma is not a fluid where DYS is physiologically present [22]. To determine if the dystrophin peptides were quantitative in plasma and to estimate the limit of detection and linearity for each tryptic peptide, we performed a calibration curve using the synthetic PrESTs. A five-parameter logistic regression model revealed goodness-of-fit (GOF) values > 0.98 for all curves with the exception of TTE (Figure 4A). All tryptic peptides had acceptable CV values below 15% for each calibration point in plasma, but CVs were only acceptable in water above 0.5 pmol/µL indicating that the limit of detection in water is above this value (Figure 4B).

Linearity is observed between 1 and 10 pmol/µL with spiked PrESTs. The lower abundance TTE tryptic peptide linearity was poor and hardly observed. Extrapolation of the intensities from control patient samples suggest that all measured patients were in the linear 1–10 pmol/µL range for YQS. When comparing with a blank water calibration curve, the YQS tryptic-peptide linearity is observed from 0.1 pmol/µL and in plasma from 1 pmol/µL. In this experimental setup the endogenous DYS cannot be distinguished until at least 1 pmol/ µL, due to the plasma matrix effect. However, our data indicate that this method is feasible for quantitation of DYS in patient samples, but further assessment is needed. The assay currently can detect DYS fragments from controls, from 3 pmol/µL, but a lower more sensitive quantification assay might be required to analyse samples from DMD patients. The assay can be improved by assessing and minimising the matrix effect from plasma. Recent advances in targeted mass spectrometry for clinical applications [35] have successfully applied an enrichment technique using antibodies to the tryptic peptides of interest which removes the majority of the matrix effect, thereby improving sensitivity. The limit of quantification in water was down to 0.5 pmol/µL, indicating that if the matrix effect could be removed the assay detection range could get down to this level.

## 3. Materials and Methods

Plasma samples from 82 human subjects were collected at the University of Ferrara (UNIFE), Italy and used to run the suspension beads immunoassay experiments. Specifically, the plasma cohort included: 3 BMD and 16 DMD patients, 16 female carriers, 33 patients affected by other neuromuscular non-DMD diseases, and 14 healthy controls. Detailed analysed subject characteristics are listed in Appendix A. Plasma from 5 human healthy donors were instead collected at the University College London (UCL, UK) and tested by targeted liquid chromatography mass spectrometry.

### 3.1. Suspension Bead Immunoassay

Two affinity purified polyclonal antibodies (HPA002725 and HPA023885) raised against different protein epitope signature tags (PrESTs) [36] were used to detect DYS. HPA002725 recognises the N-terminal domain of the protein corresponding to the region encoded between DMD exons 7 and 10. Whereas HPA023885 targets the C-terminal domain and a region between DMD exons 57 and 60. Both antibodies were produced within the Human Protein Atlas [24,37], validated by protein arrays, Western Blot and Immunohistochemical staining [38,39]. In addition, an antibody targeting the C-terminal of carbonic anhydrase 3 [13], produced and validated in-house using the Human Protein Atlas antibody production pipeline [39] was used as a positive control. All antibodies were coupled to fluorescently labelled magnetic microspheres as previously described [40]. In short, 1.75 μg of antibody was coupled to 5 × 10^6^ carboxylated MagPlex beads (Luminex corp. Austin, TX, US) using EDC/NHS chemistry and blocked in 5% (*w/v*) bovine serum albumin faction V in PBS containing 0.05% (*v/v*) Tween 20 (PBST). Successful coupling of antibodies to bead surfaces was tested through incubation with 1:2000 sheep anti-rabbit IgG R-phycoerythrin (Sigma-Aldrich, Burlington, MA, USA) in PBST for 20 min at room temperature and analysed on a Luminex LX200 system with xPONENT software (version 3.1). Coupled beads together with background control beads were pooled at equal volumes before use in immunoassay. Human plasma samples were randomised, and biotin labelled using EZ-link NHS-PEG4-Biotin (Thermo Scientific, Waltham, MA, USA) according to the manufacturers protocol. Biotin labelled plasma samples diluted to 0.07%, 0.14%, 0.28%, and 0.56% in 0.5 mg/mL rabbit IgG (Bethyl) 1:1000 ProClin^TM^ 300 (Sigma-Aldrich, Burlington, MA, USA) PBST were heat-treated for 30 min at 56 °C in a Primus 96 plus thermocycler (MWG Biotech AG, Eurofins, Ebersberg, Germany), and mixed with 2 µL suspension beads for protein capture as described previously [40]. Captured plasma proteins were detected and median florescence intensity (MFI) per individual bead ID was obtained as assay read-out using the Luminex LX200 system with xPONENT software (version 3.1). MFI signals were filtered to remove regions with a bead count below 20 (one sample removed in 0.14% plasma replicate and six samples removed in 0.07% plasma replicate). Batch effect was reduced using ComBat in R package [41] and signals were log2-transformed. Linear correlation between positive and negative control beads were used to identify and remove samples where background MFI deviated from cohort mean. Finally, mean MFI was calculated across replicates with similar plasma concentration for each remaining sample. All statistical analysis were performed in R software 4.1.0 [42].

### 3.2. Targeted Liquid Chromatography Mass Spectrometry

Two PrESTs (D1 and D2) were designed [43] and produced [44] to be used as protein standards for mass spectrometry [45]. D1 comprised 100 amino acids, corresponding to the R6-R7 spectrin repeats of the DYS Rod domain (amino acids 1017–1116, DMD exons 22 and 26, Ensemble ID: ENSG00000198947, version 108.38 assembly GRCh38.p13) whereas D2 comprised 200 amino acids corresponding to the R8-R10 residues of the DYS Rod domain (amino acids 1235-1434DMD exon 27 and 32, Ensemble ID: ENSG00000198947, version 108.38 assembly GRCh38.p13). D1 and D2 were used to develop the LC-MS/MS assay. Digested D1 and D2 peptides were obtained by trypsin’s treatment of the two PrESTs. The digested tryptic peptides were solid phase extraction (SPE) cleaned before injection into a mass spectrometry coupled with liquid chromatography. The PrESTs were spiked in plasma and used to determine the optimal transitions using Skyline software, version 20, (Macoss Lab, Washington, DC, USA) and retention times for endogenous dystrophin peptides. The digested PrESTs were also used to build a calibration curve line in water. Retention times of the D1 and D2 digested tryptic peptides are presented in Appendix A. Plasma samples (*n* = 5) were precipitated with ammonium sulphate to deplete the majority of albumin. Next, 30 µL of plasma was aliquoted with 10 µL of 30 µg/µL of the MassPREP enolase digestion standard protein (Waters Corp.) used here as a generic internal standard to control for the first sample preparation before using PrESTs. The samples were precipitated with 60 µL of 3M ammonium sulphate before being sonicated for 10 min and centrifugated at high speed for 10 min. The supernatants were taken out, and pellets were washed again with 250 µL of 1.8 M ammonium sulphate. After centrifugation, the pellets were resuspended into 40 µL digest buffer composed of 100 mM Tris, pH 7.8, 6 M Urea, 2 M Thiourea, and 2% ASB14 and mixed for 10 min. Then, 3µL of DTT solution (DL-Dithiothreitol DTT) at a concentration of 162 mM in 100 mM Tris-HCL, pH 7.8 was added. After incubation at room temperature for one hour, 6 µL of 162 mM Iodoacetamide made in 100 mM Tris-HCL, pH 7.8 was added. Samples were shaken and incubated at room temperature for one hour in the dark. A total of 330 µL of ddH_2_O was added to dilute the urea before adding 10 µL of sequencing grade modified trypsin (Promega, Madison, WI, USA) solution (1 mg/mL in 50 mM Acetic Acid). Samples were incubated for 16 h at 37 °C. Finally, digests were cleaned and desalted by SPE using a 100 mg Bond Elut C18 96 well plate (Agilent, Santa Clara, CA, USA). Samples were resuspended in 0.2% trifluoroacetic acid (TFA). Wells were primed using 1 mL of 60% Acetonitrile and 0.1% TFA followed by two equilibration washes of 1 mL of 0.1% TFA. After loading the samples, the wells were washed using 1 mL of 0.1% TFA twice. Peptides were eluted using 0.5 mL of 60% Acetonitrile 0.1% TFA and dried using a centrifugal evaporator. Samples were resuspended in 50 µL 3% Acetonitrile, 0.1% TFA before LC-MS/MS analysis.

Samples were analysed using a Xevo^TM^ TQ-XS MS coupled to an ACQUITY^TM^ UPLC^TM^ I-Class chromatography system in electrospray positive ionization (Waters corporation). Digested plasma peptides were separated using an ACQUITY^TM^ Premier Peptide BEH C18 Column, 300 Å, 1.7 µm, 2.1 × 50 mm. Mobile phases were (A) water with 0.1% formic acid and (B) acetonitrile also with 0.1% formic acid. The runs were performed over a 15 min gradient. Starting conditions were 95% A and 5% B for 5 min. Then, between 5 and 12.90 min, A decreased to 80% and B increased to 20%. From 12.90 min to 13.20, A decreased to 70% and B increased to 30%. After 13.20 min, B was held at 100% up to the end of the run. The flow rate was kept at 0.3 mL/min, with a column at 50 °C. LC-MS/MS analysis was performed using multiple reaction monitoring positive-ionization mode to detect peptides specified in Appendix A, with capillary voltage set to 2.80 kV, desolvation temperature to 400 °C and source temperature to 150 °C. Desolvation gas flow and the cone gas were kept at 600 L/Hr and 150 L/Hr respectively, nebuliser was set at 7 bars. Cone energy was set at 30 V and collision energy was variable. Data were processed using MassLynx^TM^ 4.1 and analysed using Skyline open-source software, version 20 [46]. All peptides presented in the discovery analysis in plasma are normalised to yeast enolase peptide GNPTVEVELTTEK. Calibration curves ranging from 0–100 pmol/µL were prepared in water and pooled plasma using spiked PrESTs of regions D1–D2.

## 4. Conclusions

This work is a proof-of-principle aimed to explore detection of DYS protein in plasma and the feasibility of developing a novel assay to specifically and quantitatively measure protein fragments in fluids. Quantification of DYS, if proven to be associated with DMD, could replace the non-specific CK as a DMD biomarker. In addition, such quantitative assays for blood tests might reduce the need for invasive and risky muscle biopsy procedures.

We have demonstrated the presence of DYS in human plasma using immune- and mass spectrometry-based methods. The suspension beads immunoassay suggested a variation of DYS abundance across the subjects analysed, with a significantly lower level in DMD patients compared to controls, at least for one tested antibody. The presence of DYS in human plasma from healthy donors was confirmed by LC-MS/MS. Moreover, the tryptic peptides variable abundance profiles confirmed our hypothesis that DYS fragments, instead of full DYS protein, are likely present in plasma. Additional work is needed to define the abundance profile of DYS, its fragments stability, and, more importantly, to investigate the dynamic range in DMD patients. Having a protein-specific biomarker would facilitate the assessment of determining drug efficacy in clinical trials, especially those aiming at restoring DYS protein. This would also support drug discovery for developing new therapies. Finally, these preliminary data highlight the importance of investigating more DYS domains to obtain the maximal protein coverage, to potentially identify all DMD mutations, and determine the variability of DYS expression in healthy and disease subjects to make our protein assays highly accurate and clinically feasible.

## Figures and Tables

**Figure 1 ijms-24-05215-f001:**
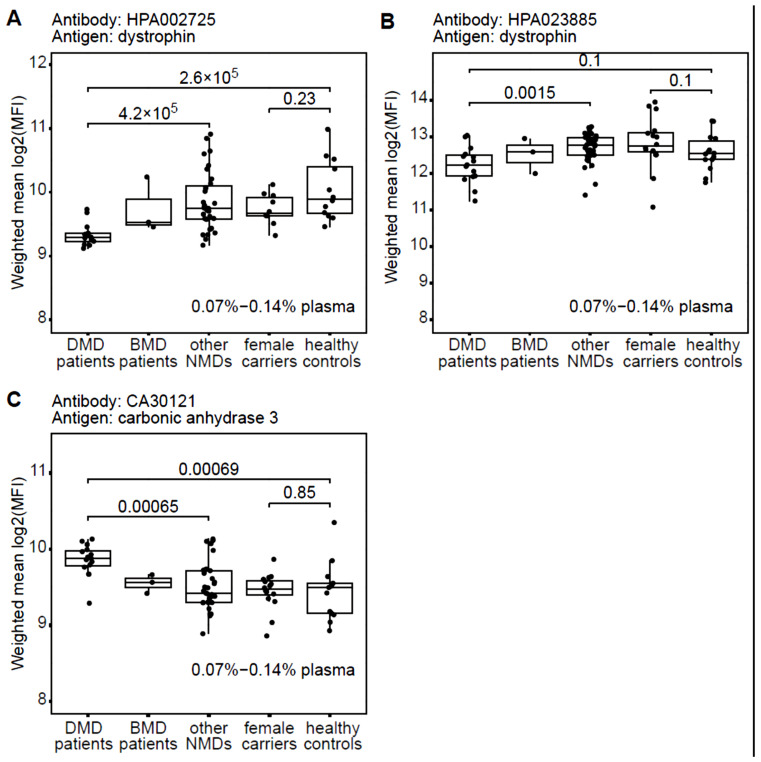
Dystrophin and carbonic anhydrase 3 detected in patient plasma cohort using antibody suspension bead array. An in-house produced antibody suspension bead array was used to detect DYS protein (**A**,**B**) and carbonic anhydrase 3 (**C**) on plasma from 82 human donors: 16 DMD patients, 3 BMD patients, 33 patients with other neuromuscular disorders not characterised by mutations within the DMD gene, as well as 16 DMD female carriers and 14 healthy controls. A significantly (*p*-value < 0.05, Wilcoxon rank sum test) lower abundance of DYS was observed in DMD patients than all other patient groups when using the N-terminally binding anti-DYS antibody HPA002725 (**A**). In contrast, the C-terminally binding anti-DYS antibody HPA023885 (**B**) showed no significant difference between DMD and healthy controls. Carbonic anhydrase 3 (**C**) was used as a positive control showing an expected higher abundance in DMD compared to healthy controls, in line with previous studies. Batch effect was compensated with the R algorithm Combat and protein abundancies calculated as mean log2(MFI) over up to 3 replicates diluted at 0.07–0.14% plasma.

**Figure 2 ijms-24-05215-f002:**
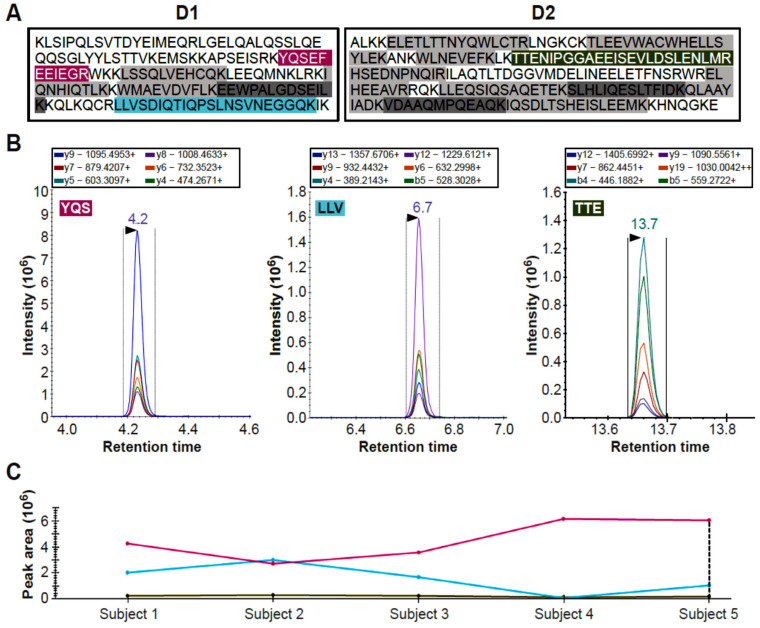
Detection of dystrophin peptides in healthy human donor plasma using targeted peptide LC-MS/MS. Two PrESTs (D1 and D2), corresponding to the R6–R7 and R8–R10 subdomains of DYS Rod domain were used to set-up a targeted mass spectrometry assay for detection in plasma from 5 healthy donors. Panel (**A**) shows the sequences for the D1 and D2 PrESTs, tryptic peptides that were developed into a multiplex targeted mass spectrometry assay. Those that were not detectable in control plasma are indicated in grey and those that were, are coloured. Panel (**B**) shows the peptide chromatograms of the mass transitions for the 3 tryptic peptide standards (YQSEFEEIEGR, LLVSDIQTIQPSLNSVNEGGQK and TTENIPGGAEEISEVLDSLENLMR) which were subsequently detected in plasma. Each coloured curve corresponds to a different transition, the colour code is maintained across the three graphs. Panel (**C**) shows summaries of the peak area across 5 healthy donor samples. The TTE tryptic peptide has the lowest intensity and area, followed by the LLV peptide. On the other hand, the YQS peptide has much better intensity. Lines have the same colours as the labelled peptides in Panel (**A**).

**Figure 3 ijms-24-05215-f003:**
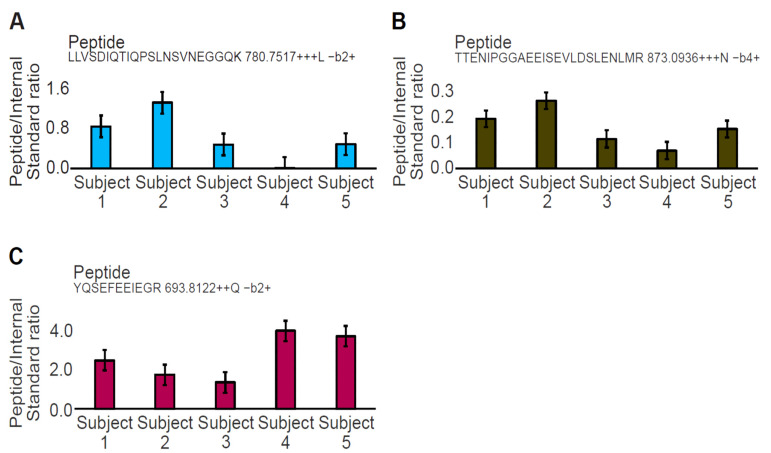
Quantitation of the three DMD tryptic peptides in plasma across 5 healthy control samples. Graphs (**A**,**B**) are related to the tryptic peptides LLV and TTE of the D1 (R6–R7) and the D2 PrESTs (R8–R10), respectively. Graph (**C**) is related to the tryptic peptide YQS of the D1 PrESTs (R6–R7). The 3 tryptic peptides have different expression profiles across each subject and do not correlate. Transition information used for quantitation is given with the peptide sequence for each graph.

**Figure 4 ijms-24-05215-f004:**
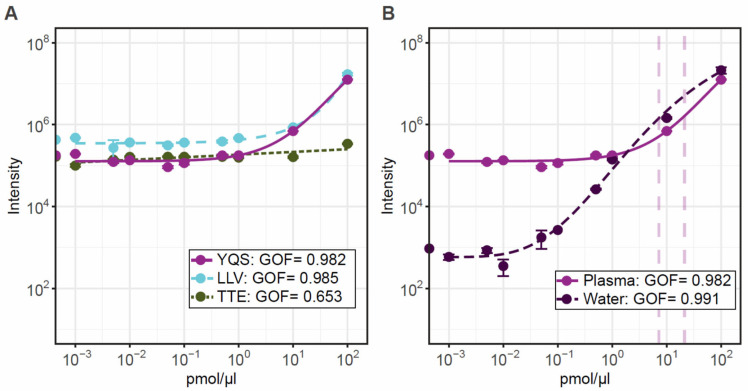
Calibration curves for the three peptides calculated using a 5-parameter logistic regression model. Panel (**A**) shows the standard curves of increasing peptide concentration over a wide range of 0.001–100 pmol/µL for each peptide detectable in control plasma. Panel (**B**) shows the standard curve of the optimal peptide YQS in plasma and water to show the matrix effect from plasma. Graphs are displayed with log10 axis, and each data point represents triplicate values.

## Data Availability

Data is contained within the article and Appendix A.

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
