# Peer review of "A Proof of Principle Proteomic Study Detects Dystrophin in Human Plasma: Implications in DMD Diagnosis and Clinical Monitoring"

_ijms, 2023, doi:10.3390/ijms24065215_

Round 1

Reviewer 1 Report

Dear authors,

Please find below my comments concerning the manuscript entitled « A proof-of-principle proteomic study detects dystrophin in human 2 plasma: implications in DMD diagnosis and clinical monitoring ».

Rossi et al. describe here a study that aims to determine the ability to detect the dystrophin protein in human plasma. The ultimate goal of the approach is to have a powerful biomarker for the diagnostic screening of DMD and the monitoring of the impact of therapies. To do this, the authors successively tested i) an immunological approach with two antibodies specific to fragments of the protein on samples from patients with DMD or other neuromuscular diseases, female carriers and healthy subjects and ii) a targeted liquid chromatography mass spectrometry approach on samples from healthy individuals. They show a reduction of the dystrophin signal in plasma samples from DMD patients compared to healthy subjects with one antibody and the detection of three peptides in plasma from healthy donors.

The manuscript is overall well-constructed and written. The introduction section clearly sets the context of the study and explains the scientific objective with regard to the current limitations encountered in supporting the diagnosis of the disease in the face of new variants, in examining the phenotype/genotype correlation, in limiting the use of muscle biopsy or in reducing the time and cost of analysis of outcomes measures. The rationale of the study is clearly of interest and the question asked to define DMD biomarker is important. The experimental design is suitable and detailed. The results are analyzed according to adequate statistical rules and well presented. Nevertheless, we can regret that there is great variability in the values determined in the different groups or plasma samples, whether in western blotting or mass spectrometry, which raises questions about the biological significance and the use of the data to identify a relevant biomarker. Despite this experimental limit, the results already provide informative findings on the possibility to quantify dystrophin in plasma and thus constitutes a proof-of-principle as mentioned by the authors.

In regard of the interest of the question asked on the management of patients and the overall quality of the manuscript, I consider that the manuscript could be recommended for publication if the authors are able to answer the few points highlighted below.

Major points

- My main concern is the disparity of the values obtained within groups of individuals, which unfortunately suggests a certain degree of heterogeneity. In order to have a more accurate reading of the data produced by western blotting and thus to rule on the relevance of the experimental approach, it seems appropriate to provide additional information on individuals (age, gender) included in the study, to group the 33 non-DMD patients according to their disease and if possible to integrate these variables into the analysis. It is intriguing to see such divergent MFI values in controls (some of which are equivalent to those obtained in patients), which questions the discriminative character of the HPA002725 antibody in particular. The authors should discuss this heterogeneity. Similarly, knowing that CA3 accumulation is associated with damage of type 1 muscle fiber, one wonders about the sensitivity of the antibody with rather low MFIs between DMD patient samples and control subjects or even an absence of difference.

- Another point is that the results section lacks the presentation of the blots to visualize the bands and profiles obtained between the groups.

Minor points

- As much as the usefulness of dystrophin protein detection is clear for diagnostic screening of DMD and evaluation of the impact of novel therapies, its use to help DMD monitoring is more complicated to perceive. Therefore, it would be better to delete " to aid disease monitoring " in the abstract.

- There is an inconsistency between the comments in the text of the Result section “Tryptic peptides LLV and TTE are the lowest abundant peptides (figure 3 A and B), and they share the same expression profile with the only exception that LLV is not detected in one sample (figure 3A). In contrast, the most abundant YQS tryptic peptide showed a different expression profile than LLV and TTE, among subjects (figure 3C).” and in the legend of Figure 2 “The TTE tryptic peptide has the lowest intensity and area and shows the same expression pattern for all samples. The YQS and LLV peptides have much better intensity and show variability in their abundance across samples.”, which needs to be corrected.

- Lines 317-318. “The lack of concordance observed between peptide abundance, could be due to an individual sample effect (ie lipemia or administered drugs).” To be clarified because not very comprehensible.

- Typing errors:

Line 311(Figure3 legend). Delete "healthy" before plasma

Line 318. Replace “ie” by “i.e.”

Lines 326, 338, 341 and 346. Add a space in front of pmol/ul

Reviewer 2 Report

Duchenne muscular dystrophy (DMD), is the most common mono genetic disease, affecting 1 in 3500 live male births. To date and as pointed out by the authors, creatine kinase analysis is the only routinely used biomarkers for screening of DMD. This lacks specificity and often does not correlate with the severity of the disease. An alternative and more accurate diagnostic assay would require a invasive muscle biopsy and is not only time consuming, but not very cost effective as well. Indeed, there is a need for a better diagnostic screening for DMD. Here, the authors reported a novel assay, detecting dystrophin protein fragments in human plasma by a suspension bead immunoassay using two validated anti-dystrophin specific antibodies. They also demonstrated the detection of dystrophin protein using an antibody-independent method (targeted liquid chromatography mass spectrometry).  While these methods need to be validate and conducted in larger sample cohorts, this may be the future for an improved diagnostic screening and clinical monitoring of DMD patients.

Minor correction:

Fig 1; pg 7: Please enlarge the font size for axis titles, axis labels and tick label

Fig 2B and 2 C, pg 8. Please enlarge font size for axis title and labels 

Fig 3, pg 9: Please enlarge the font size for all graphs (Asix labels, titles and tick). 

Reviewer 3 Report

In the presented manuscript authors showed a proof of principle study to detect the dystrophin protein (or fragments of) in plasma using antibody dependent and independent methods. Authors presented with the need to find biomarkers for diagnosis of DMD using less invasive diagnostic techniques. The concepts/techniques used in paper advances the field. However, I have few concerns regarding the experiment design and some results and explanation for those

1.  Since the authors did not see the significance with c-terminal specific antibody (HPA023885), have they considered using other c-terminal specific antibodies against dystrophin? 

2. The mean MFI for HPA023885 was ~10 fold higher as compared to the other two antibodies as shown in supplemental figure 1. Was the antibody optimized? In the methods section, authors mentioned 1.75 ug of antibody was used to coat the beads. Did they try to reduce that concentration to bring the assay in the linear range of detection?

3. The peptides were detected in the control sample that were collected from other source. This portion should be explored more in the patient samples and can be a full manuscript by itself. 

Apart from this, there are some minor revisions needed as well.

1. Line 371 Supplementary Materials: Did not mention supplementary figure 2

2. The supplementary figures do not have captions and/or lack figure labels  

3. The quality of figure 2 should be improved. It is hard to read the labels.

Round 2

Reviewer 3 Report

The authors have answered all the concerns and emphasize that this is proof of concept study. The experiments suggested are part of their future manuscript.